# Pathophysiology and Clinical Manifestations of COVID-19-Related Acute Kidney Injury—The Current State of Knowledge and Future Perspectives

**DOI:** 10.3390/ijms22137082

**Published:** 2021-06-30

**Authors:** Iwona Smarz-Widelska, Ewelina Grywalska, Izabela Morawska, Alicja Forma, Adam Michalski, Sebastian Mertowski, Rafał Hrynkiewicz, Paulina Niedźwiedzka-Rystwej, Izabela Korona-Glowniak, Miłosz Parczewski, Wojciech Załuska

**Affiliations:** 1Department of Nephrology, Cardinal Stefan Wyszynski Provincial Hospital in Lublin, 20-718 Lublin, Poland; i.widelska@interia.pl; 2Department of Clinical Immunology and Immunotherapy, Medical University of Lublin, 20-093 Lublin, Poland; izabelamorawska19@gmail.com (I.M.); adam.96.michalski@gmail.com (A.M.); mertowskisebastian@gmail.com (S.M.); 3Department of Forensic Medicine, Medical University of Lublin, 20-090 Lublin, Poland; aforma@onet.pl; 4Institute of Biology, University of Szczecin, 71-412 Szczecin, Poland; rafal.hrynkiewicz@usz.edu.pl; 5Department of Pharmaceutical Microbiology, Medical University of Lublin, Chodźki 1, 20-093 Lublin, Poland; iza.glowniak@umlub.pl; 6Department of Infectious, Tropical Diseases and Immune Deficiency, Pomeranian Medical University in Szczecin, 71-455 Szczecin, Poland; mparczewski@yahoo.co.uk; 7Department of Nephrology, Medical University of Lublin, 20-954 Lublin, Poland; wojciech.zaluska@umlub.pl

**Keywords:** COVID-19, SARS-CoV-2, acute kidney injury, AKI, survival rate, glomerulopathy, SARS-CoV-2 tropism, coronavirus

## Abstract

The continually evolving severe acute respiratory syndrome coronavirus 2 (SARS-CoV-2) pandemic has resulted in a vast number of either acute or chronic medical impairments of a pathophysiology that is not yet fully understood. SARS-CoV-2 tropism for the organs is associated with bilateral organ cross-talks as well as targeted dysfunctions, among which acute kidney injury (AKI) seems to be highly prevalent in infected patients. The need for efficient management of COVID-related AKI patients is an aspect that is still being investigated by nephrologists; however, another reason for concern is a disturbingly high proportion of various types of kidney dysfunctions in patients who have recovered from COVID-19. Even though the clinical picture of AKI and COVID-related AKI seems to be quite similar, it must be considered that regarding the latter, little is known about both the optimal management and long-term consequences. These discrepancies raise an urgent need for further research aimed at evaluating the molecular mechanisms associated with SARS-CoV-2-induced kidney damage as well as standardized management of COVID-related AKI patients. The following review presents a comprehensive and most-recent insight into the pathophysiology, clinical manifestations, recommended patient management, treatment strategies, and post-mortem findings in patients with COVID-related AKI.

## 1. Introduction

Coronavirus disease 2019 (COVID-19), induced by severe acute respiratory syndrome coronavirus-2 (SARS-CoV-2), is a fairly recent disease, first reported less than 2 years ago. According to John Hopkins Coronavirus Resource Center, it has been responsible for more than 2 million deaths as of the 16 March 2021 [1]. Patients diagnosed with COVID-19 present a wide range of various impairments and organ dysfunctions, while the course of the disease depends on the individual, ranging from asymptomatic infection to severe symptoms and even fatal incidents [2,3]. The most common clinical manifestations include interstitial and alveolar pneumonia, with around 5–20% progressing to acute respiratory distress syndrome (ARDS), septic shock, or multiple organ failure [3,4,5,6]. Despite the most prevalently diagnosed respiratory manifestations, gastrointestinal, neurological, olfactory, and cardiac symptoms are also very common in patients of various ages. Additionally, there is plausible evidence from the literature suggesting that COVID-19 might be a risk factor for an acute kidney injury (AKI), with its incidence being estimated from around 4.8 to 36% in COVID-19 patients [7,8,9,10,11,12,13,14]. This incidence may vary in different ethnic groups [15]. Published data demonstrate that approximately 30% of patients hospitalized with COVID-19 will develop AKI, while such risk is significantly increased in critically ill patients [16,17,18]. According to a recent meta-analysis, approximately 5.4% of COVID-19 patients require renal replacement therapy (RRT), with this number increasing up to 16.4% among patients admitted to intensive care units (ICUs) [8]. COVID-related AKI is associated with poor outcomes of the disease and greater mortality rates [19,20,21], reaching 11.1–14.4% compared with 1.2–5% among infected non-AKI patients [10,22]. Although the number of patients with COVID-19-associated AKI is greater than that in other sepsis-related conditions, the rate of AKI in COVID-19 patients is proportional to that observed in other forms of sepsis [23]. (Figure 1)

It is worth mentioning that from the practical experience of the authors of the manuscript (MP), approximately 80% of hospital-admitted COVID-19 patients present with the clinical symptoms of dehydration (due to, e.g., fever or hyperventilation). It has become a practical routine that virtually every in-hospital-admitted patient receives 1000–2000 mL of intravenous fluids/day in the first days following admittance. Therefore, it should be underlined that dehydration may interact with and/or exacerbate the pathological processes contributing to renal failure.

The above-mentioned data indicate an urgent need for an understanding of the pathophysiology of renal injury caused by SARS-CoV-2 infection. In this review, we aimed to describe the most recent knowledge regarding the pathophysiology, clinical manifestations, and proposed management strategies of COVID-19 patients who develop AKI in the course of the infection.

## 2. Acute Kidney Injury—Definition and Diagnostic Criteria

Acute kidney injury (AKI) is commonly diagnosed in critically ill patients with its occurrence estimated at up to 50% in patients hospitalized in ICUs [24]. A rapid increase in serum creatinine levels or decrease in urine output lies at the foundation of the condition [25]. AKI is a growing healthcare problem and is associated with high morbidity and mortality rates among patients. Moreover, the treatment of this syndrome requires a long hospitalization course and is generating high healthcare costs due to its cost-ineffective nature, especially in the times of the SARS-CoV-2 pandemic [25,26]. According to the Kidney Disease Improving Global Guidelines (KDIGO) 2012, clinical criteria of this condition are implemented both from the Acute Kidney Injury Network (AKIN) criteria and RIFLE criteria [27,28], as the diagnosis of AKI can be established in one of the following clinical situations: an increase in serum creatinine levels ≥0.3 mg/dl within 48 h; an increase ≥1.5 times baseline within the previous 7 days; or urine volume ≤0.5 mL/kg/h for 6 h [29]. The Acute Disease Quality Initiative (ADQI) Group recommends the use of biomarkers in the process of diagnosis of surgery-associated AKI [30]. The proposed biomarkers include tissue damage markers such as TIMP2/IGFBP7 (combination of tissue inhibitor of metalloproteinases 2 and insulin-like growth factor binding protein) and NGAL (neutrophil gelatinase-associated lipocalin) in patients at high risk of surgery-associated AKI. The appropriatness of the use of these biomarkers in COVID-19 patietns has been evaluated by two authors. In the paper by Luther et al., the urine levels of renal tissue damage markers (TIMP2, NGAL, and KIM-2) in COVID-19 patients that developped AKI were elevated [31]. Similarly, He et al. demonstrated that a high NGAL urine level is a predictor of AKI and of high risk of in-hospital death [32]. Together, these findings show that the incorporation of such biomarker evaluation in COVID-19 patients represents an additional tool for the identification of patients at high risk of AKI development.

The term ‘acute’ in relation to kidney disease means that the symptoms of functional or structural damage last no more than 90 days [33,34]. As previously mentioned, AKI can be caused by various health conditions, mostly hepato- and cardio-renal diseases [35,36], sepsis [37], and bacterial toxins [38]; moreover, it frequently overlaps with other chronic or acute renal diseases or may be diagnosed in patients without any coexisting health problems [39]. Anatomically, with reference to kidneys, AKI can be divided into prerenal, intrinsic, and postrenal syndrome [25]. KDIGO proposed staging of AKI based on the laboratory parameters (serum creatinine level/eGFR) and urine output [39] (Table 1). Typically, it develops without any specific symptoms and alarming signs, and, therefore, it can remain unnoticed for a long time, especially if the patient does not come to a specialist ward in a life-threatening condition [25]. It is necessary to be extremely vigilant not to overlook the correct diagnosis. 

## 3. Mechanisms of COVID-19-Associated AKI

The pathogenesis of COVID-related AKI is multifactorial and includes direct and indirect mechanisms of viral infection; both of them remain poorly understood and require further evaluation (Figure 2 and Table 2). Although some of the pathological findings are similar to those of other types of AKI, researchers have described mechanisms typically involved in COVID-related AKI. Most of the mechanisms overlap with each other, enhancing their intensity; therefore, it is difficult to clearly distinguish them.

The pathological changes in kidney during COVID-19-associated AKI include tubulointerstitial, glomerular, and vascular damage.

The kidney picture presents with diffuse proximal tubule injury with loss of the brush border and frank necrosis accompanied by vacuolar degeneration and tubulointerstitial fibrosis. Electron microscopy imaging shows SARS-CoV-2 viruses in the tubular epithelium (predominantly in the proximal tubule and podocytes) [3,40]. In the interstitial compartment, inflammatory cell forms infiltrate, and edema can be seen. In the case of sever kidney injury, the basement membrane is the only barrier between the filtrate and the peritubular interstitium. Because of the increased endothelial permeability, glomerular filtrate leaks from the tubular lumen into the interstitium [40].

In the glomeruli, the observation of autopsies from kidneys of COVID-19 patients shows the diffuse and focal segmental fibrin thrombus in the glomerular capillary loops and endothelial injury. In the case of collapsing glomerulopathy, glomerular epithelial damage occurs together with loss of podocytes integrity. Glomerular capillaries are segmental or globally collapsed and sclerotic, with hyperplasia and hypertrophy of the glomerular epithelium. Some cases present with diffuse erythrocyte stagnation in the glomerular capillary or glomerular loop occlusion by erythrocytes over peritubular capillaries [40].

On the vascular level, the picture of AKI demonstrates vasoconstriction of intrarenal vessels, increased vascular permeability, and microthrombi formation [3]. Vascular endothelium damage occurs, which can be observed as swelling of endothelial cells. The leukocyte–endothelium interactions are enhanced, leading to leukocyte migration into the interstitium [40,41].

### 3.1. Sites of SARS-CoV-2 Invasion

SARS-CoV-2 enters the renal cells using the receptor angiotensin-1 converting enzyme (ACE2), and for the endocytosis and membrane fusion, it uses transmembrane serine protease 2 (TMPRSS2). The S1 domain of SARS-CoV-2 (protein S1) mediates the entry of the virus into target host cells after binding with the transmembrane ACE2 receptor [42]. The virus has marked tropism for the respiratory tract, bowel, and heart cells, where ACE2 expression is well represented. In the kidney, ACE2 is present in podocytes, proximal tubule, mesangial cells, parietal epithelium of Bowman’s capsule, and collecting ducts [43,44,45,46]. Importantly, ACE2 is expressed in the kidney stronger than in the lungs. The virus may enter the kidney by invading podocytes; then, in the tubular fluid, it accesses ACE2 in the proximal tubule [47]. It was hypothesized that ACE2 gene polymorphisms may predispose to kidney injury induced by SARS-CoV-2 infection [48,49,50]. Interestingly, SARS-CoV-2 infection both prevents ACE2 from attaching to the receptor and alternates ACE2 expression within the proximal tubular cells, especially in the areas of acute tubular injury [3,51]. Accumulation of the AGII protein, not converted to AG1-7, promotes inflammation by increasing cytokine release and allowing easier immune cell infiltration of the tissues [52,53].

Another receptor with a potential impact on the entry of SARS-CoV-2 is neuropilin-2 (NRP-1) [54]. NRP-1 is implicated in several aspects of a SARS-CoV-2 infection, including possible spread through the olfactory bulb and into the central nervous system and increased NRP-1 RNA expression in lungs of severe COVID-19 [54]. Moreover, NRP-1 may also serve as an immune checkpoint of the memory T cell in COVID-19 [55]. It has been identified that binding of the SARS-CoV-2 spike protein to the NRP-1 receptors impacts the docking of the VEGF-A ligand [55]. NRP-1 may act as a host cell mediator that is able to increase the infectivity of the virus and thus contribute to the tropism of the coronavirus [56]. Studies on the potential use of NPR-1 as a target for therapy are ongoing [57]. It is worth noting that the role of NRP-1 has also been confirmed in kidney diseases via elevated vascular permeability and endothelial cell apoptosis leading to imbalanced kidney regulations also increasing renal disorders [58].

In addition, studies have confirmed that priming the spike proteins of SARS-CoV-2 may also be dependent on transmembrane protease serine 2 (TMPRSS2) and other proteases, e.g., furin [56].

### 3.2. Renin–Angiotensin–Aldosterone System Impairment by SARS-CoV-2

The renin–angiotensin–aldosterone system (RAAS) plays a vital role in the maintenance of proper arterial pressure and tissue perfusion; therefore, any dysfunctions within this system take part in the pathogenesis of renal diseases.

SARS-CoV-2 infects cells using ACE2 as its receptor. ACE2, through the modulation of the RAAS, plays an important physiologic role in the homeostasis of tissue microcirculation and inflammation [59]. It converts angiotensin II (AGII) to angiotensin 1-7 (AG1-7), a protein with anti-inflammatory, vasodilatory, anti-fibrotic, and natriuretic activity [60,61]. AG1-7 exerts this activity by binding to a G-protein-coupled Mas receptor (MasR). In contrast to AG1-7, AGII presents pro-inflammatory, pro-fibrotic, and anti-diuretic properties. It acts also as a vasoconstrictor [52]. All of these actions are mediated by binding AGII to the Ang T1 receptor (AT1R) on the cellular surface [59]. Interestingly, opposite effects may be induced if AGII attaches to Ang T2 receptors (AT2Rs). These two receptors for AGII are differentially distributed in the renal tissue. AT1R is predominately expressed in the cortex and AT2R in the medulla, which results in distinct AGII effects in these two renal localizations: vasoconstriction at the renal cortex and vasodilation at the medulla. SARS-CoV-2 infection unbalances the RAAS since cellular internalization of viruses induces degradation of the membranal ACE2. This results in depletion of Ang1-7, which may lead to microcirculatory dysfunction, enhanced inflammatory processes, hypercoagulability, fibrosis, and tissue damage.

### 3.3. Viral-Induced Tissue Damage

One of the possible causes of AKI in COVID-19 patients is tissue damage directly caused by the renal tropism of SARS-CoV-2; however, the scientific data on this matter remain unclear.

In 2019, a researcher from Wuhan, China, described the presence of the SARS-CoV-2 genetic material using the RT-PCR method in urine samples and suspected that the new coronavirus may also be involved in renal failure [62]. Some studies have proved acute kidney tubular injury in COVID-19 patients as well as the presence of virions in these organs in post-mortem examination [63,64,65].

However, these observations remain unconfirmed in some studies. For example, Rabb et al. described tubular injury in the renal biopsy samples of critically ill patients with SARS-CoV-2 but without detection of the viral genetic material from the renal tissue, urine, and serum samples [66]. Moreover, Sharma et al. did not find viral material from renal biopsy despite the fact that the patients were diagnosed with COVID-related AKI and all had tubular injury [67]. It is worth noting that the differences between those results may also be the cause of the different stage of virus replication. The authors also emphasize that acute renal tubular damage is associated with ischemia rather than viral damage, during which the glomeruli or interstitial tissue is damaged [68]. Interestingly, post-mortem observations also revealed pathogenic changes in the renal glomeruli in the form of sclerotic capillary vessels and renal epithelial hyperplasia [69]. Computed tomography (CT) scans of the kidneys in COVID-19 patients show the signs of edema and ongoing renal inflammation [19]. Direct failure of renal tissue has also been proven using microscopy by describing the presence of SARS-CoV-2-like particles in damaged podocytes, which were also detached from the glomerular membrane [63].

Patients with COVID-related AKI can develop collapsing glomerulopathy, an aggressive variant of focal segmental glomerulonephritis, which is typically caused by various viral infections in both direct and immune-mediated mechanisms (Figure 1) [70,71,72,73,74]. There is a described genetic (high-risk *APOL-1* gene) and ethnic (Africans) susceptibility to COVID-19-associated glomerular disease [64,73,75]. The immune particles, such as macrophages and complement components C5b-C9, triggered by the virus, also influence the pathogenesis of COVID-related AKI by promoting innate immune-related cell damage at the tubulointerstitium [76].

Nevertheless, many pathologists believe that the presence of the viral-like particles may be misleading and that it does not clearly indicate the presence of replicating SARS CoV-2 in given tissues [77,78].

### 3.4. Ischemia

Renal ischemia in the course of COVID-19 may result from septic shock, which is an extremely dangerous complication of this disease. It is estimated that septic shock may develop in up to 6% of all patients with SARS-CoV-2 infection, and it is the key cause of multiorgan failure and a third major cause of death in COVID-19 patients [12,79,80]. The main pathological mechanism associated with kidney damage in septic shock is hypoperfusion resulting in tissue hypoxia and ischemia [3]. Renal ischemia is one of the main causes of tubular damage. It rapidly induces a number of structural and functional alterations in renal proximal tubular epithelial cells, which can be described as acute tubular injury [3,41]. The tubular epithelial cells undergo injury and, if it is severe, death by apoptosis and necrosis (acute tubular necrosis), with organ functional impairment of water, electrolyte homeostasis, and reduced excretion of waste products of metabolism [40].

The assumption of an ischemic background of AKI in COVID-19 is supported by the numerous pathologies found in post-mortem studies: erythrocyte-related occlusion of glomerular capillary loops and peritubular small vessels, proximal tubule injury with necrosis, and degeneration of vacuoles [3,63]. As previously mentioned, the mechanisms overlap, and the dysregulation within the RAAS system in the pathogenesis of renal ischemia in COVID-related AKI is also reported. The elevated level of AGII resulting from ACE2 degradation by SARS-CoV-2 and subsequent AG1-7 depletion may predispose to vasoconstriction in renal vessels [41,80]. Local blood flow to the outer medulla, reduced due to arteriolar vasoconstriction, is further compromised by local edema [40].

Ischemic complications can also be induced indirectly by respiratory failure. Lung damage commonly observed in patients with severe COVID-19 leads to generalized hypoxemia and hypercapnia. Renal tissue is physiologically extremely well supplied in blood; thus, a decrease in blood oxygenation and its reduced supply, together with the increased demand caused by sepsis, increases the risk of ischemic damage [81]. Alveolar damage in the lungs has been reported to be a cause of damage in renal mesangium, endothelium, glomeruli, tubules, and the accumulation of white blood cells in renal tissue [82]. Moreover, hypercapnia is associated with neurohormonal dysregulation, with increased activity of the sympathetic part of a nervous system with its vasoconstrictive potential [83].

Interestingly, it was reported that life-saving mechanical ventilation has a negative impact on kidney oxygenation and is associated with a significantly higher risk of AKI development. Mechanical ventilation causes many physical and physiological changes in lung tissue and in the respiratory muscles functioning, which results in reduced cardiac output and exacerbation of renal edema, reducing the flow of the oxygenated blood through the kidneys [82,84].

### 3.5. Rhabdomyolysis

Rhabdomyolysis, a rare condition where damaged skeletal muscles break down, has been reported as a complication of severe COVID-19 [85,86]. Rhabdomyolysis may contribute to kidney injuries occurring in COVID-19 patients [86]. Myoglobin derived from damaged skeletal muscles induces renal injury in several mechanisms. Myoglobin itself forms pigment casts in the tubules obstructing the normal flow of fluid. Moreover, direct tubulotoxicity is mediated by reactive oxygen species (ROS) generated by iron released from heme. In the third mechanism, medullar vasoconstriction exacerbates intrarenal hypoxia, leading to renal failure [87]. In post-mortem investigations of 26 COVID-19 patients, pigmented casts have been detected in kidney tubules. The clinical history of these patients has shown elevated serum creatine kinase levels, which is a marker of muscle damage. The etiology of both findings remains unknown; however, they are probably associated with rhabdomyolysis [63]. A case series report presenting data from renal biopsies of 10 COVID-19 patients who had clinical features of AKI also demonstrated myoglobin casts in renal tubules together with elevated serum creatine kinase levels, both probably due to the ongoing rhabdomyolysis [67].

### 3.6. Thrombotic Events

There are multiple observations of thrombotic events in patients with COVID-19, even when they are under anticoagulation treatment [79,88,89]. In most patients with SARS-CoV-2, elevated levels of D-dimers were observed and were correlated with an increased risk of renal failure [12,62,79,90]. Moreover, higher mortality rates were observed in the group of people with COVID-19 and increased thrombotic risk [89]. Furthermore, post-mortem examinations of kidney tissues indicated the presence of fibrin thrombi in renal vessels, which suggests that thrombotic complications are considered one of the causes of COVID-related AKI. Sharma et al. described the thrombotic microangiopathy in two patients with SARS-CoV-2-induced AKI; however, both were in the at-risk group because of their disorders [67]. Venous thrombosis as a cause of renal failure has also been described by Xia et al. [68].

### 3.7. Hyperinflammation

Multiple overlapping mechanisms, as described earlier, can lead to hyperinflammation. The cytokine release syndrome (CRS) and further hyperinflammatory state are some of the major pathological mechanisms described in COVID-19 and hence in COVID-related AKI. Viral replication itself was reported to be associated with the induction of a hyperinflammatory state [91]. Patients with confirmed infection present significantly higher plasma levels of inflammatory markers and factors than those of uninfected patients due to the onset of the cytokine storm, which is associated with a more severe course of the disease and greater morbidity rates in such patients. One of the major pathways associated with the induction of hyperinflammation includes the enhanced activation of the complement system and subsequent production of C3a and C5a responsible for the initiation of the pro-inflammatory responses [92]. Other relevant features include the impairments within CD4+ CD8+ T cells and NK cells in addition to the overactivation of macrophages and neutrophils [93,94,95]. There is debate regarding whether pharmacological immunomodulation should be implemented in patients with COVID-related AKI.

### 3.8. Drug-Induced AKI

Many drugs with a potential nephrotoxic effect are used to improve the clinical outcome of patients with the SARS-CoV-2, including those in the critical condition. Reports from post-mortem investigations in patients who developed AKI during COVID-19 showed the presence of crystals in the proximal kidney tubules and casts; these results may support the hypothesis [68]. Moreover, in two case reports, drug-related kidney damage in the form of oxalate nephropathy was described [96].

## 4. Laboratory Examinations and Histopathology

### 4.1. Laboratory Findings

Among the basic parameters determined in patients with suspected kidney injury and the acute condition are the level of serum creatinine and the eGFR index. In the case of COVID-related AKI, as in other AKI cases, serum creatinine levels are much higher compared to those of healthy individuals, and, of course, the eGFR level is significantly reduced; in almost 70% of patients, several abnormalities in the urine samples are detected and include proteinuria, hematuria, and, less frequently, leukocyturia [19,97,98]. Most patients with COVID-related AKI present an increased white blood cell count with elevated levels of neutrophils and a lowered number of lymphocytes [68]. Selected laboratory parameters reflect an increased AKI risk in the course of the COVID-19 disease.

Elevated IL-6 activity was linked with a higher risk of renal failure. Moreover, patients with elevated levels of IL-6, creatinine and D-dimers are those with significantly greater mortality rates [68,99]. The observations of laboratory abnormalities in severe COVID-19 coincide with each other in the following aspects: severely ill patients were reported to present higher levels of IL-6, D-dimers, LDH, platelets, hsCRP, white blood cells with predominant neutrophils, and fewer lymphocytes [62,100,101,102,103,104,105,106]. While high levels of platelets are usually described, lowered platelet levels may be associated with a higher risk of death [107,108,109]. In addition, researchers showed that the presence of protein in urine upon admission to hospital was associated with a higher risk of developing AKI, and in patients with SARS-CoV-2 who subsequently developed proteinuria, it was associated with a higher risk of death. In addition, patients who presented hematuria more frequently required admission to the ICU, needed mechanical respiratory assistance, and had a higher risk of death [110]. Laboratory findings on patients with COVID-related AKI are summarized in Table 3.

### 4.2. Histopathologic Features

At the very beginning of the pandemic, the heterogenous guidelines regarding the handling of the patients who died due to COVID-19 were significantly restricted in order to prevent potential post-mortem viral transmission resulting in limited knowledge about the macroscopic and microscopic changes occurring during COVID-19. Nevertheless, the introduction of proper personal protective equipment (PPE) allowed for further investigations, enabling the study of microscopic damages induced by SARS-CoV-2 infection and, thus, the exact pathophysiological mechanisms of viral action. COVID-19 patients who suffer from AKI present a wide spectrum of various glomerular and tubular impairments [14]. Acute tubular injury remains the most common constituent of AKI; thus, its clinical (both macroscopic and microscopic) components are primarily observed during either ante- or post-mortem studies [111]. Interestingly, the severity of the acute tubular injury was demonstrated to be associated with creatinine levels [112]. Other less common microscopic pathologies include acute endothelial injury, collapsing glomerulopathy, acute tubular necrosis, podocytopathy, acute interstitial nephritis, and microangiopathy [67,113,114].

Su et al. in a post-mortem study on 26 COVID-19 patients with AKI confirmed the presence of viral particles in the kidneys, especially in the cytoplasm of renal proximal tubular epithelium, distal tubules, and podocytes [63]. The observations by electron microscopy showed increased thickness of the glomerular basement membrane, expansion of the mesangium, and microvillous transformation. Even the light microscopy showed significant SARS-CoV-2-induced damages, including proximal acute tubule injury, vacuolar degeneration, the obstruction of glomerular capillary loops by erythrocytes, deposits of hemosiderin in the tubular epithelium, and even necrosis. Furthermore, antemortem studies presented the cytopathic effects exerted by SARS-CoV-2 on the cells of the proximal tubules and podocytes [115]. During the autopsy of those with either confirmed or suspected COVID-related AKI, both the renal and myocardial tissues were recommended to be sampled for further studies [15]. Postmortem studies demonstrated that the most common characteristics of acute tubular injury include vacuolization, cytoplasmic simplification, luminal ectasia, and either degenerative or regenerative nuclear changes [112]. While the above-mentioned data remain poorly described in the literature, they provide plausible evidence of kidney SARS-CoV-2 tropism [116].

## 5. Management and Treatment Options

Since the ongoing pandemic is still evolving while the pathomechanisms of SARS-CoV-2 infection are continually discovered, there are clear management guidelines for COVID-related comorbidities such as AKI. To date, there are no supportive data that show an advantage of a particular treatment over another; in fact, there is no specific treatment for COVID-19 AKI. The implementation of the supportive care guidelines included in the Kidney Disease: Improving Global Outcomes (KDIGO) is recommended in COVID-19 AKI patient management [117]. All COVID-19 patients administered to the hospital who are supposed to be at a high risk of AKI development should be monitored in terms of potential kidney injury; close monitoring of creatinine levels along with urine output is recommended [118]. Except for the COVID-19 infection itself, AKI might occur as a result of the donation of infected organs. The risk of acquiring COVID-19 from organs dedicated for donation is very low; however, it was demonstrated that SARS-CoV-2 presents a significant tropism for the kidney, and, thus, a potential infection from a donor is possible [119]. Therefore, the major approaches to proper management of COVID-19 AKI patients include the control of the potential source of infection, proper isolation of a patient, hemodynamic support, electrolyte (especially hyperkaliemia) and acid-base control, fluid balance, avoidance of potentially nephrotoxic drugs, and asymptomatic supportive care [120].

The involvement of extracorporeal therapies is of major importance due to the presence of bilateral damages, such as the coexisting tubular–alveolar, lung–kidney, or cardiovascular–kidney crosstalk [121]. Approximately, in up to 31% of cases of critically ill patients, clinicians decide to introduce RRT; however, the above-mentioned aspects, such as potentially infected organs from donations, should be taken into consideration [122]. Moreover, while considering RRT, clinicians should be aware of the increased risk of coagulopathy in AKI patients with COVID-19 [123].

Some antiviral drugs are currently used to treat COVID-19-related AKI. Remdesivir (nucleotide analog inhibiting viral RNA-dependent RNA polymerase) was reported to be effective as an initial treatment for patients with AKI and without a concomitant liver disease; however, it is registered for use in patients with eGFR>30 mL/min per 1.73 m^2^ [124]. The administration of remdesivir and convalescent plasma has been approved by the US Food and Drug Administration (FDA) as one of the potential COVID-19 treatment strategies. A viral drug with a similar mechanism of action—favipiravir—has also been administered to COVID-19 AKI patients; however, its efficacy is unconfirmed [125,126]. Since one of the characteristics of AKI is a cytokine storm, tocilizumab—a humanized monoclonal antibody to IL-6 receptor—might be potentially used in COVID-19 AKI patients [127]. Another proposed therapy includes the administration of interferons; however, their efficacy to date has been poor [128]. There is also strong evidence of a positive effect of hemodialysis in patients with COVID-19-induced AKI [129].

Attention should be drawn to the potential therapeutic approach based on restoration of the impaired RAAS balance during COVID-19 infection [130]. It is suggested that the cytokine storm and the coagulopathy during COVID-19 infection may be linked to the depletion of ACE2 and AG1-7, which, in the conditions of homeostasis, has a visible renoprotective impact [130]. Stimulation of the ACE2/AG1-7/MasR axis may be a potential target for COVID-19-associated AKI treatment.

Due to the lack of rigorous restrictions regarding COVID-19 AKI treatment, clinicians should be aware that the benefits of the proposed treatment must outweigh the possible negative effects, and each therapy should be individualized for every patient.

## 6. Clinical Outcome and Survival Rate

Despite the introduction of several types of COVID-19 vaccines and their proven efficacy in the prevention of the spread of SARS-CoV-2 infection, morbidity and mortality rates remain high in the general population. Similarly, those rates are unpredictable in cases of underlying malignancies and medical conditions, such as AKI, due to impaired immunological responses as a consequence of infection [131,132]. Moreover, it seems difficult to estimate the mortality risk of COVID-19 patients because of new reports continuously suggesting previously unknown pathophysiological mechanisms of SARS-CoV-2 infection in addition to a wide spectrum of immunological responses depending on the patient’s clinical condition and concomitant disorders. The duration, as well as the recovery rates of COVID-19 AKI patients, remains unknown [133]. However, the presence of COVID-induced AKI in hospitalized patients is significantly associated with worse clinical outcomes and higher mortality rates compared to those of patients without this complication [99,118,119,134,135,136,137,138]. Furthermore, AKI related to SARS-CoV-2 infection is associated with greater mortality rates compared to those of AKI induced by other causes, and its prevalence increases with the severity of infection [129,134,139,140]. Interestingly, it was observed that chronic treatment with ACE inhibitors or angiotensin receptor blockers (ARBs) might result in a significantly greater risk of AKI in patients with SARS-CoV-2 via upregulated ACE2 levels [141]. On the other hand, ARBs or ACE inhibitors may restore the RAAS balance impaired by SARS-CoV-2 invasion. Depleting AGII or blocking its receptor AT1R would enhance the ACE2/AG1-7/MasR axis exerting anti-inflammatory, anti-fibrotic, and vasodilating properties [59]. Therefore, despite the unclear effects of RAAS inhibitors on ACE2 levels, on susceptibility for SARS-CoV-2 infection, and on COVID-19 severity, RAAS inhibitors should be continued in COVID-19 patients or those at risk of it [142].

Several researchers reported that the mortality rate of COVID-19 AKI patients varies from 35 to 80% and might increase when RRT necessarily reaches the range of 75–90% [18,143,144]. It was demonstrated that advanced AKI stages or its progression, as well as the older age of patients and medical history of heart failure, constitute potential risk factors associated with higher mortality rates; interestingly, there was no such association in the case of patients who required RRT [99,139,145]. Except for the above-mentioned factors, diabetes and hypertension along with increased baseline serum creatinine levels and increased serum IL-6 levels are associated with a higher risk of COVID-19-related AKI; stage 3 AKI (according to KDIGO) is an independent predictor of death in those patients [68,146]. Despite a high mortality percentage, some patients were reported to recover up to three weeks after the onset of AKI symptoms [147]. A reason for concern is that, apart from the unsatisfactory survival rates, those who survived COVID-19 and the related AKI present a high proportion of persistent kidney dysfunctions [11,148]. Therefore, the control of renal functions of all COVID-19 patients (especially plasma creatinine) is crucial to minimize further damages and prevent potential disabilities [149]. It is clinically important to further investigate the possible factors associated with worse clinical outcomes of COVID-19 patients with AKI. For those patients who have tested positive for SARS-CoV-2 infection, it is recommended to provide a consultation with a nephrologist to minimize the potential side effects of possible COVID-19-induced AKI, providing better clinical outcomes for those patients at the same time.

## 7. Future Perspectives

One of the major aspects associated with a comprehensive understanding of the pathophysiology of AKI during the course of COVID-19 includes the need for more renal biopsies to be performed. Data obtained during the microscopic examinations would significantly help us to understand the process more comprehensively, providing an ability to identify the potential risk factors and establish guidelines for the prevention of AKI. Large, multicenter studies are required to fully understand not only the impact of COVID-19 on kidney dysfunctions during the course of infection but also its long-term effects, as well assisting in the search for non-invasive ways of identifying patients with an increased risk of AKI. Moreover, there is a need for further continuation of the clinical trials focused on both newly identified drugs and previously known medications in order to introduce the most effective treatment options.

## 8. Conclusions

Kidney injury associated with the course of COVID-19 can be either caused by multiorgan failure or directly induced by the SARS-CoV-2 kidney tropism. The primary findings in renal biopsy are acute tubular injury and epithelial necrosis, but SARS-CoV-2 infection may exacerbate preexisting kidney conditions, such as lupus nephritis or membranous glomerulopathy. Furthermore, studies suggest that the pathophysiology of COVID-19-related AKI is similar to that of sepsis-associated AKI, as the majority of damage within the kidneys is due to the hemodynamic and immunologic effects of the infection, as supported by the increased levels of CRP and Il-6. Moreover, the increased levels of D-dimers and fibrinogen were found in the blood samples of patients diagnosed with COVID-19-related AKI; since the increase in the levels of the above-mentioned factors correlates with the severity of kidney damage, it might indicate a thrombotic mechanism of renal injury. The intensity of ACE2 receptor expression in patients diagnosed with COVID-19 remains debatable, with some studies indicating overexpression and others its downregulation. A plethora of drugs is currently under investigation in relation to their utility to treat COVID-19-related AKI. Remdesivir was observed to be effective in such treatment. Additionally, because of the cytokine storm etiology of AKI, tocilizumab showed promising results in clinical trials. AKI is associated with a higher rate of negative outcomes; thus, although the incidence of AKI during the course COVID-19 is significantly lower compared to that of ARDS, there is a growing demand for an understanding of its pathophysiology and standardized patient management in order to reduce morbidity and mortality rates.

## Figures and Tables

**Figure 1 ijms-22-07082-f001:**
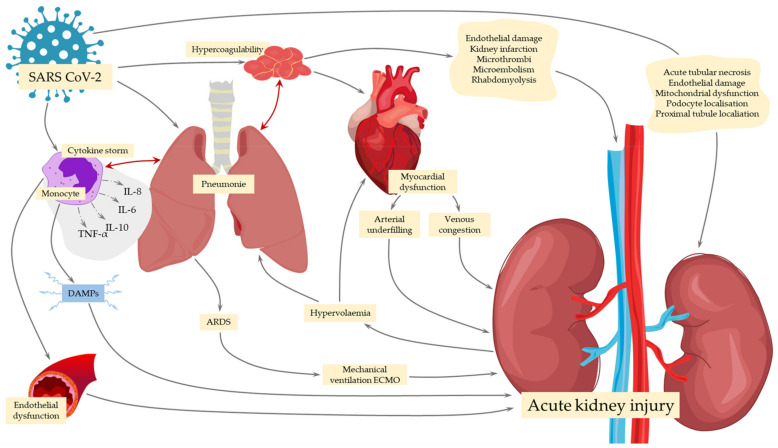
The consequences of SARS CoV-2 infection leading to AKI. SARS CoV-2, severe acute respiratory syndrome coronavirus 2; IL, interleukin; ARDS, acute respiratory distress syndrome; DAMPs, damage-associated molecular patterns; ECMO, extracorporeal membrane oxygenation; TNF-α, tumor necrosis factor-alpha.

**Figure 2 ijms-22-07082-f002:**
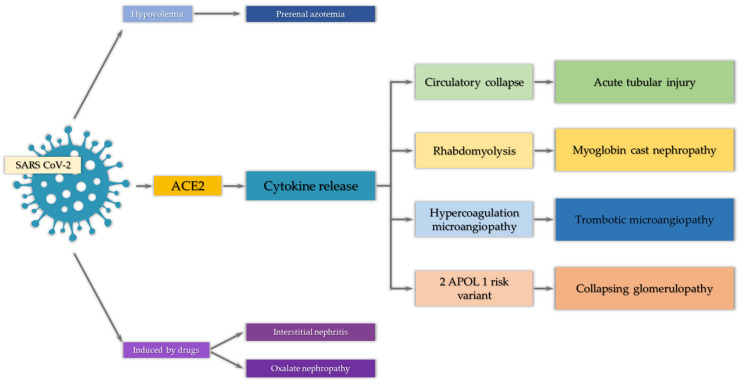
The summary of COVID-related AKI. SARS CoV-2, severe acute respiratory syndrome coronavirus 2; ACE2, angiotensin-converting enzyme 2; APOL1, apolipoprotein L1.

**Table 1 ijms-22-07082-t001:** Staging of AKI according to the Kidney Disease Improving Global Guidelines [39].

Stage	Creatinine Level/Estimated Glomerular Filtration Rate (eGFR)	Urine Output
1	1.5–1.9 times baseline or ≥0.3 mg/dL (≥26.5 mol/L) increase	<0.5 mL/kg/h for 6–12 h
2	2.0–2.9 times baseline	<0.5 mL/kg/h for ≥12 h
3	3 times baseline; ≥4.0 mg/dL (≥353.6 mol/L) increase; initiation of replacement renal therapy (RRT); or in patients <18 years, a decrease in eGFR<35 mL/min/1.73 m^2^	<0.3 mL/kg/h for ≥24 h or anuria ≥12 h

**Table 2 ijms-22-07082-t002:** Potential mechanisms of kidney damage associated with the SARS-CoV-2 infection.

Components of SARS-CoV-2 Infection	Mechanisms of Kidney Damage
Cytopathic effect of replicating virus	Immune-mediated tissue damage
	Glomerulonephropaty
	Tubular injury
Renin–angiotensin–aldosterone system abnormalities	Proinflammatory properties of upregulated AGII
	Polymorphism in ACE2
	Upregulation/downregulation of ACE2
Ischemic injury	Septic shock and inflammation
	Hypoxia
	Hypercapnia
	Mechanical ventilation
Rhabdomyolysis	Medullary vasoconstriction, intrarenal hypoxia, and damage of renal cells by reactive oxygen species
Microthrombi	Inflammatory infiltrations, obstruction, and subsequent hypoxia
Hyperinflammation	Cytokine storm
	Complement activation

**Table 3 ijms-22-07082-t003:** Laboratory findings on patients with COVID-related AKI.

Material	Abnormal Parameter	Elevated/Lowered
Urine samples	Protein	Elevated
	Hematuria	
Blood count	WBC	Elevated
	Neutrophils	
	Platelets	
	Lymphocytes	Lowered
Pro-inflammatory markers	Ferritin	Elevated
	IL-2R	
	IL-6	
	hsCRP	
	LDH	
Coagulation markers	D-dimer	Elevated
Renal markers	Creatinine	Elevated
	Blood urea nitrogen	
	eGFR	Lowered

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
