# Peer review of "Pathophysiology and Clinical Manifestations of COVID-19-Related Acute Kidney Injury—The Current State of Knowledge and Future Perspectives"

_ijms, 2021, doi:10.3390/ijms22137082_

Round 1
Reviewer 1 Report
In the article “Pathophysiology and clinical manifestations of COVID-19-related acute kidney injury – the current state of knowledge and future perspectives” by Iwona Smarz-Widelska et al., the authors aim to provide a comprehensive insight into the pathophysiology, clinical manifestations, recommended patient management, treatment strategies, and post-mortem findings in patients with COVID-related AKI. I congratulate the authors for successfully completing this review article. However, there are some issues that should be addressed to improve the quality of the manuscript.
Major concern:
1) In addition to its primary host receptor ACE2, other host receptors such as neuropilin-1 may also play pivotal roles in COVID-19 and contribute to kidney injury. For a better understanding of the pathophysiology of COVID-related AKI, a paragraph discussing such a host receptor would benefit this article. (Recommended references: NRP1 in COVID-19: PMID: 33395426; PMID: 33815686. NRP1 in kidney disease: PMID: 33744890. Possible drug target: PMID: 33787218)
Minor concerns:
1) There could be some modifications to improve the quality of Figure 1, for example, arrows of aetiologies involving renal vessels should be pointed at the renal artery or vein.
2) In Table 2, the mechanism of rhabdomyolysis-induced kidney damage might be more complicated than the prevention of renal blood flow.
3) It could be better to integrate Figure 2 with Table 2, as they both focus on the mechanism of SARS-CoV-2 infection-induced kidney injury.
Author Response
Dear Reviewer 1,
Thank you very much for your time and valuable comments towards our manuscript. We have corrected the manuscript accordingly, and here are point-by-point responses to your concerns:
Major concern:
1) In addition to its primary host receptor ACE2, other host receptors such as neuropilin-1 may also play pivotal roles in COVID-19 and contribute to kidney injury. For a better understanding of the pathophysiology of COVID-related AKI, a paragraph discussing such a host receptor would benefit this article. (Recommended references: NRP1 in COVID-19: PMID: 33395426; PMID: 33815686. NRP1 in kidney disease: PMID: 33744890. Possible drug target: PMID: 33787218)
We have added the data on the NRP-1 receptor in the Section 3.1. Sites of SARS-CoV-2 invasion, as suggested by the Reviewer (lines 172-186).
Minor concerns:
1) There could be some modifications to improve the quality of Figure 1, for example, arrows of aetiologies involving renal vessels should be pointed at the renal artery or vein.
We have corrected the Fig.1 according to the Reviewer’s suggestion.
2) In Table 2, the mechanism of rhabdomyolysis-induced kidney damage might be more complicated than the prevention of renal blood flow.
We have added additional mechanisms of kidney damage in rhabdomyolysis-induced kidney damage in the Table 2
3) It could be better to integrate Figure 2 with Table 2, as they both focus on the mechanism of SARS-CoV-2 infection-induced kidney injury.
We would rather like to leave the Fig.2 and Tab.2 separately, as we expanded the Figure with additional elements.
We hope that our manuscript in its current form will fulfill the requirements. Again, thank you very much for your time and consideration.
On behalf of the Authors,
Paulina Niedźwiedzka-Rystwej
Reviewer 2 Report
A comprehensive review on AKI, my complaints. Tables are readable and graphics are attractive. Only minor revisions are required to enlarge the spectrum of information.
-If the authors have practical experience with Covid-19 patients should highlight the patient's tendency toward dehydration due to fever, NIV, hyperventilation, etc. In our hospital, patients with feeding problems or frail were hydrated up to 40 cc/kg/day. A pre-renal etiology is quite probable, without excluding cytokine ad viral etiology. Hence, if the authors agree, a practical note on the risk of dehydration could be added.
-The authors should add few words about ACE (no about ACE-inhibitors). I advise two reviews:
The Role of the Renin-Angiotensin System in Severe Acute Respiratory Syndrome-CoV-2 Infection; DOI: https://doi.org/10.1159/000507914
Kidney and Lung ACE2 Expression after an ACE Inhibitor or an Ang II Receptor Blocker: Implications for COVID-19; DOI:https://doi.org/10.1681/ASN.2020050667
Last suggestion: the interesting paper "COVID-19 and AKI: Where Do We Stand?
DOI: https://doi.org/10.1681/ASN.2020121768" explains why there is a difference in AKI rate.
Author Response
Dear Reviewer 2,
Thank you very much for your time and valuable comments towards our manuscript. We have corrected the manuscript accordingly, and here are point-by-point responses to your concerns:
If the authors have practical experience with Covid-19 patients should highlight the patient's tendency toward dehydration due to fever, NIV, hyperventilation, etc. In our hospital, patients with feeding problems or frail were hydrated up to 40 cc/kg/day. A pre-renal etiology is quite probable, without excluding cytokine ad viral etiology. Hence, if the authors agree, a practical note on the risk of dehydration could be added.
The appropriate paragraph is added in the Introduction (lines 68-74).
The authors should add few words about ACE (no about ACE-inhibitors). I advise two reviews:
The Role of the Renin-Angiotensin System in Severe Acute Respiratory Syndrome-CoV-2 Infection; DOI: https://doi.org/10.1159/000507914
Kidney and Lung ACE2 Expression after an ACE Inhibitor or an Ang II Receptor Blocker: Implications for COVID-19; DOI:https://doi.org/10.1681/ASN.2020050667
We have added these two references to the section 3.1. Sites of SARS-CoV-2 invasion (lines 160,163); information about ACE has been expanded.
Last suggestion: the interesting paper "COVID-19 and AKI: Where Do We Stand?
DOI: https://doi.org/10.1681/ASN.2020121768" explains why there is a difference in AKI rate.
This reference has been added in the Introduction, line 67.
We hope that after the corrections and rearrangements, the manuscript will fulfill the requirements.
Thank you again for your time and consideration.
On behalf of the Authors,
Paulina Niedźwiedzka-Rystwej
Reviewer 3 Report
This is an interesting highly relevant but complicated topic, providing many references. Yet, the manuscript is confusing, unorganized, and ignores principal pathophysiologic derangements. A thorough structural reorganization and re-writing is needed.
Comments:
Lines 95 and further are out of context in this section 2, and should perhaps be moved to the introduction, or form another section altogether;
As suggested by the ADQI team (for instance Nadim, J Am Heart Assoc. 2018 Jun 1;7(11):e008834), you should address biomarkers in section 2 dealing with the definition of AKI. Surprisingly, this has not been extensively studied in COVID, patients, with few exceptions (Luther, Acta Anaesthesiol Scand. 2021 Mar;65(3):364-372; He, Kidney Dis (Basel). 2021 Mar;7(2):120-130). The incorporation of renal biomarkers in the assessment of AKI in COVID-19 disease, seemingly, is of supreme importance.
Section 3 heading (line 105): should be modified, perhaps to something like "Mechanisms of COVID-19-associated AKI"
Renal morphology of COVID-19-associated AKI, currently incorporated in subsection 3.1, deserves a separate section, perhaps preceding the current Section 3. Do address separately glomerular, vascular and tubulointerstitial changes. Subsequently, in the original section 3.1 you may address viral presence or absence, and so on regarding morphologic findings relevant to the other pathophysiologic mechanisms.
Noteworthy, when addressing the possibility of direct renal viral invasion I believe you should mention the presence and distribution of viral acceptors on renal cells, namely ACE2.
Figure 1 is needed but is totally misleading: (1) Collapsing glomerulopathy has nothing to do downstream to rhabdo; (2) Thrombotic microangiopathy is not specifically derived from circulatory collapse; (3) ATN is likely derived from various combined mechanisms, including hypoxia/ischemia related to systemic and renal microcircular injury and coagulopathy, inflammation and oxidative stress, perhaps direct viral invasion, the impact of rhabdo (that acts through direct tubulotoxicity mediatred by ROS), altered renal microcirculation (likely related to increased NO scavenging and increased interstitial pressure) and tubular obstruction, etc.; (4) what about SIRS, cardio-renal and pulmonary-renal syndromes?; (5) what about glomerulopathy?
Section 3.2 does not truly address viral-induced tissue damage, as the title indicates, but rather deals with viral attachment to cells expressing ACE2. This important topic should be placed very early in the manuscript, along with the concept of the outcomes of the imbalanced ACE/Ang II/At1R and ACE2/Ang-(1-7)/MasR axes in COVID-19 disease (a crucial topic which is only marginally and insufficiently addressed in this review. Please look at Abassi, Front Physiol 11: 574753, 2020, regarding the balance between these pressor and depressor arms of RAS, which have opposing effects on inflammation, vascular tone, oxygenation, coagulation, cell death and fibrosis. This fundamental topic might be moved upstream to a renewed section 3
Section 3.3 – ischemia. There is a confusing mix here. This topic is highly relevant for the tobulointerstital component, with its high oxygen consumption for tubular transport, particularly in the outer medulla. Renal parenchymal ischemia in COVID-19 disease indeed likely represent ACE2 depletion, leading to unbalanced pressor and depressor arms of RAS. In addition, as partially explained, systemic hypoxia, microangiopathy, and increased renal interstitial pressure further compromise medullary oxygenation. By contrast, glomerular injury has nothing to do with ischemia whatsoever.
Section 3.4 – Rhabdo occurs and likely does take a place in COVID-associated AKI, but as mentioned above, the mechanisms are much more compound than simply pigment casts (See Heyman, J Am Soc. Nephrol 7:1066-1074, 1996;
Sections 3.8, 3.9 – again, absolutely misplaced, underscoring the need for a total re-organization of the manuscript. They are not parts of the mechanisms involved, but rather a description of the phenotype
Section 4 – overall one of the few focused sections. I suggest addressing the conceptual approach of restoration of the impaired RAS balance (Heyman, J Clin Med 10(6): 1200, 2021)
Section 5, regarding reference 142 and the use of ACEi/ARBs – again, look at the imbalanced RAS related to reduced ACE2/Ang 1-7/MasR axis (Abassi, Front Physiol 11: 574753, 2020), the rationale for maintaining ACEi/ARBs (Vaduganatham, March 30, NEJM 2020)
Author Response
Dear Reviewer 3,
Thank you very much for your time and valuable comments towards our manuscript. We have corrected the manuscript accordingly, and here are point-by-point responses to your concerns:
Comments:
Lines 95 and further are out of context in this section 2, and should perhaps be moved to the introduction, or form another section altogether;
These lines have been moved to the Introduction (lines 55-64).
As suggested by the ADQI team (for instance Nadim, J Am Heart Assoc. 2018 Jun 1;7(11):e008834), you should address biomarkers in section 2 dealing with the definition of AKI. Surprisingly, this has not been extensively studied in COVID, patients, with few exceptions (Luther, Acta Anaesthesiol Scand. 2021 Mar;65(3):364-372; He, Kidney Dis (Basel). 2021 Mar;7(2):120-130). The incorporation of renal biomarkers in the assessment of AKI in COVID-19 disease, seemingly, is of supreme importance.
Appropriate paragraph has been added to the section 2. Acute Kidney Injury – Definition and Diagnostic Criteria, lines 100-111. Suggested papers have been referenced.
Section 3 heading (line 105): should be modified, perhaps to something like "Mechanisms of COVID-19-associated AKI"
The heading has been changed as suggested by the Reviewer.
Renal morphology of COVID-19-associated AKI, currently incorporated in subsection 3.1, deserves a separate section, perhaps preceding the current Section 3. Do address separately glomerular, vascular and tubulointerstitial changes. Subsequently, in the original section 3.1 you may address viral presence or absence, and so on regarding morphologic findings relevant to the other pathophysiologic mechanisms.
Renal morphology of COVID-19-associated AKI has been described in the section 3. Mechanisms of COVID-19-associated AKI (lines 132-153).
Noteworthy, when addressing the possibility of direct renal viral invasion I believe you should mention the presence and distribution of viral acceptors on renal cells, namely ACE2.
This issue is addressed in the section 3.1 Sites of SARS-CoV-2 invasion.
Figure 1 is needed but is totally misleading: (1) Collapsing glomerulopathy has nothing to do downstream to rhabdo; (2) Thrombotic microangiopathy is not specifically derived from circulatory collapse; (3) ATN is likely derived from various combined mechanisms, including hypoxia/ischemia related to systemic and renal microcircular injury and coagulopathy, inflammation and oxidative stress, perhaps direct viral invasion, the impact of rhabdo (that acts through direct tubulotoxicity mediatred by ROS), altered renal microcirculation (likely related to increased NO scavenging and increased interstitial pressure) and tubular obstruction, etc.; (4) what about SIRS, cardio-renal and pulmonary-renal syndromes?; (5) what about glomerulopathy?
Thank you for your suggestion. There is a misleading formulation of the Figure 2 (we suposed the Reviewer had Figure 2 in mind while writing this comment) and we have corrected it and added additional information.
Section 3.2 does not truly address viral-induced tissue damage, as the title indicates, but rather deals with viral attachment to cells expressing ACE2. This important topic should be placed very early in the manuscript, along with the concept of the outcomes of the imbalanced ACE/Ang II/At1R and ACE2/Ang-(1-7)/MasR axes in COVID-19 disease (a crucial topic which is only marginally and insufficiently addressed in this review. Please look at Abassi, Front Physiol 11: 574753, 2020, regarding the balance between these pressor and depressor arms of RAS, which have opposing effects on inflammation, vascular tone, oxygenation, coagulation, cell death and fibrosis. This fundamental topic might be moved upstream to a renewed section 3.
Section 3.2 is now entitled 3.2. Renin-angiotensin-aldosterone system impairment by SARS-CoV-2. RAAS balance and ACE/Ang II/At1R and ACE2/Ang-(1-7)/MasR axes have been described. Paper by Abassi has been referenced.
Section 3.3 – ischemia. There is a confusing mix here. This topic is highly relevant for the tobulointerstital component, with its high oxygen consumption for tubular transport, particularly in the outer medulla. Renal parenchymal ischemia in COVID-19 disease indeed likely represent ACE2 depletion, leading to unbalanced pressor and depressor arms of RAS. In addition, as partially explained, systemic hypoxia, microangiopathy, and increased renal interstitial pressure further compromise medullary oxygenation. By contrast, glomerular injury has nothing to do with ischemia whatsoever.
This section is now numbered 3.4. Your suggestions have been implemented.
Section 3.4 – Rhabdo occurs and likely does take a place in COVID-associated AKI, but as mentioned above, the mechanisms are much more compound than simply pigment casts (See Heyman, J Am Soc. Nephrol 7:1066-1074, 1996;
The section has been expanded (lines 289-295); current number of the section is 3.4.
Sections 3.8, 3.9 – again, absolutely misplaced, underscoring the need for a total re-organization of the manuscript. They are not parts of the mechanisms involved, but rather a description of the phenotype.
Both sections (current number 4.1 and 4.2) are moved to the new chapter 4. Laboratory examinations and histopathology.
Section 4 – overall one of the few focused sections. I suggest addressing the conceptual approach of restoration of the impaired RAS balance (Heyman, J Clin Med 10(6): 1200, 2021)
The paragraf introducing data on the RAS balance has been referenced, as suggested by the Reviewer (lines 437-442).
Section 5, regarding reference 142 and the use of ACEi/ARBs – again, look at the imbalanced RAS related to reduced ACE2/Ang 1-7/MasR axis (Abassi, Front Physiol 11: 574753, 2020), the rationale for maintaining ACEi/ARBs (Vaduganatham, March 30, NEJM 2020)
The data from the suggested papers have been referenced (lines 464-469).
We hope that the manuscript in its current form, after the corrections, will fulfill the requirements.
Once again, thank You very much for your time and consideration.
On behalf of the Authors,
Paulina Niedźwiedzka-Rystwej